# When Blood Disorders Meet Cancer: Uncovering the Oncogenic Landscape of Sickle Cell Disease

**DOI:** 10.3390/jcm14238509

**Published:** 2025-11-30

**Authors:** Elise Casadessus, Manon Saby, Stéphanie Forté, Yves Pastore, Vincent-Philippe Lavallée, Thomas Pincez

**Affiliations:** 1Division of Pediatric Hematology-Oncology, CHU Sainte-Justine, 3175 Chemin de la Côte Sainte-Catherine, Montreal, QC H3T 1C5, Canada; 2Faculty of Medicine, Université de Montréal, Montreal, QC H3T 1J4, Canada; 3CHU Sainte-Justine Azrieli Research Center, Montreal, QC H3T 1C5, Canada; 4Centre de Recherche du Centre Hospitalier de l’Université de Montréal (CRCHUM), Montreal, QC H2X 0A9, Canada

**Keywords:** sickle cell disease, malignancy, leukemia, hydroxyurea, clonal hematopoiesis, hematopoietic stem cell transplantation

## Abstract

Sickle cell disease (SCD) is a hemoglobinopathy characterized by hemolysis, vaso-occlusion, and systemic inflammation. Epidemiological studies identified an increased risk of leukemia, especially acute myeloid leukemia (AML), in individuals with SCD, whereas data regarding other tumors are conflicting. SCD-associated AMLs frequently display high-risk features with unfavorable karyotypes and a dismal prognosis. SCD is associated with multiple phenomena linked to carcinogenesis in other contexts, including chronic inflammation, oxidative stress, ineffective erythropoiesis, accelerated hematopoietic aging, impaired tumor immunosurveillance, and increased clonal hematopoiesis. The role and respective contribution of these disease-intrinsic mechanisms in SCD remain to be studied. Although therapies used in SCD could theoretically modulate the risk of malignancies, no data exist to support an increased or reduced risk associated with their use. The most notable exception is hematopoietic stem cell transplantation and, to a lesser extent, gene therapy, for which the conditioning and/or procedure itself is known to increase the risk of leukemia. In sum, the effect of SCD on carcinogenesis is an emerging area of investigation with data supporting specificities in SCD-associated AML. Future research is required to determine the role of treatments to mitigate the increased risk and improve the outcome of SCD-associated AML.

## 1. Introduction

Sickle cell disease (SCD) is an autosomal recessive hemoglobinopathy caused by a single nucleotide variation in the sixth codon of the β-globin gene, substituting glutamic acid with valine [1]. This variation results in the production of abnormal hemoglobin, hemoglobin S (HbS), which tends to polymerize while deoxygenated. The intraerythrocytic polymerization-depolymerization cycle alters red blood cells (RBCs), modifying their conformation, rheological properties, and adhesion characteristics, and ultimately leading to their destruction through hemolysis. SCD is associated with a wide range of physiological alterations, including chronic inflammation, oxidative stress, procoagulant state, neutrophilic activation, and endothelial dysfunction [2]. Evidence also supports the existence of ineffective erythropoiesis and several features evocative of premature aging [3,4].

SCD is associated with a wide range of acute and chronic complications that can affect virtually every organ [2]. In addition to pain crisis, the hallmark of the disease, patients with SCD may notably experience acute chest syndrome, stroke, and early organ failure.

SCD is estimated to be the twelfth leading cause of death worldwide in children under 5 [5]. With the current optimized management available, the life expectancy of patients with SCD is reduced by ~20 years [2,6]. The advances in care made in recent decades have markedly improved life expectancy, thereby unmasking age-related complications, such as bone disease, pulmonary hypertension, and neurocognitive impairment [7,8], but also malignancies. This report of malignancies, mainly leukemia, was somewhat surprising. Although the coincidental occurrence of leukemia in individuals with SCD was expected, clinical features of these leukemia and epidemiological data suggested that SCD could be an independent risk factor for leukemia occurrence. The recent development of potentially curative approaches for SCD has also been marked by the occurrence of hematological malignancies (leukemia and myelodysplastic syndrome [MDS]) following hematopoietic stem cell transplantation (HSCT) or gene therapy. The underlying causes of a potential increased risk of leukemia were difficult to reconcile with the conceptualization of SCD as a congenital disorder of mature RBCs. Progress in SCD understanding uncovered the dynamic alterations present in SCD, and progress in biology identified the potential leukemogenesis consequences of these alterations. This emerging evidence paves the way for a better understanding of this risk and the mechanisms, including the effect of the underlying treatment. The potential cancer risk associated with SCD is poorly known, and this novel research field uncovered an important aspect of this disease. This cancer risk could be modified by the treatment received and could, by itself, impact treatment consideration.

This review aims to synthesize current knowledge on the oncogenic landscape of SCD, with a particular focus on the incidence and characteristics of leukemia in this population, whether arising spontaneously or following therapeutic interventions, and to explore the potential biological mechanisms linking SCD to increased cancer susceptibility.

## 2. Epidemiological Studies of Cancer Risk in Sickle Cell Disease

Several pieces of evidence suggest that individuals with SCD may have an increased risk of cancer, particularly hematological malignancies such as acute leukemias. The first data point came from a single-institution study that estimated the overall cancer incidence in this population at 1.74 cases per 1000 patient-years [9]. From the 696 patients studied, one developed a non-Hodgkin lymphoma and four developed a non-hematologic tumor. Conversely, a subsequent multicenter study including 16,613 patients and two epidemiological studies suggested that the risk of solid tumors was only marginally increased and that hematological malignancies and especially acute leukemias were notably overrepresented in individuals with SCD [10,11,12]. The two epidemiological studies used population databases to compare individuals with and without SCD [13]. The study by Brunson et al. (n = 6423 patients for 141,752 person-years) found a 72% increased rate of leukemia and specifically acute myeloid leukemia (AML) (standardized incidence rate = 3.59) in the Californian population. This risk appeared to be increased from the age of 15 [12]. The increased risk of chronic lymphocytic leukemia was borderline, and the study found a 32% reduced risk of solid tumors. A study by Seminog et al. (n = 7512 patients) in England confirmed the increased risk of leukemia, and especially AML (rate ratio = 10) [11]. However, this study also found a higher rate ratio for lymphoma and most solid tumors. One notable exception was the risk of breast cancer, which was lower, as in the study from California (Table 1). There was no trend toward an increased risk after the approval of hydroxyurea in California, suggesting that SCD itself may predispose individuals to leukemogenesis [12]. The multicenter retrospective study from Origa et al. included both individuals with thalassemia and SCD from eight Italian centers [14]. The study found that the most frequent tumor was a liver tumor and did not report leukemia in the 815 individuals with SCD.

Thus, the two epidemiological studies are concordant toward an increased risk of leukemia, especially AML, despite its precise magnitude remaining to be clarified. While both studies found a reduction in breast cancer, data regarding other solid tumors are discrepant. As these studies did not report the characteristics of leukemia, they did not analyze whether there was specificity in leukemia occurring in individuals with SCD.

## 3. Clinical Features of Leukemia Reported in Sickle Cell Disease

Since 1972, 53 cases of acute leukemia in patients with SCD have been reported in the literature (Table 2). The male-to-female ratio was 0.75, and the median age at diagnosis was 26 years (range: 3–61). In total, 13 patients (25%) had acute lymphoblastic leukemia (ALL), 1 had undifferentiated acute leukemia, and 39 (73%) had myeloid neoplasms, including 17 AML, 6 MDS, and 16 overlapping MDS/AML. Among the 16 patients diagnosed with MDS/AML and with data available, 8 (50%) had 10–20% of blasts, suggesting that the AML occurred from a pre-existing MDS.

Among the 27 patients with available cytogenetic or molecular data, 2 ALL cases were Philadelphia-positive, 1 AML had a normal karyotype, 3 had acute promyelocytic leukemia, and 20 patients with MDS and/or AML showed unfavorable cytogenetics such as −5, −7, del(17), 11q23 rearrangement, and chromosome 3 alterations and/or mutations in TP53, KMT2A, RAS, RUNX1, or PTPN11 (Table 3). According to the 2022 ICC classification [15], 55% of reported SCD-associated AML fulfilled the criteria for AML with myelodysplasia-related or therapy-related features, frequently involving −7/del(7q), −5/del(5q), TP53 mutations, or complex karyotypes. Only a minority corresponded to de novo AML or acute promyelocytic leukemia with PML::RARA.

Most patients (84%) were homozygous for HbSS, 8% had HbSC, 6% HbSβ^0^-thalassemia, and 1 patient had HbSD-Los Angeles (Table 4). Hydroxyurea exposure was documented in 41 patients: 17 (45%) had never received hydroxyurea, while 24 (55%) had been treated prior to acute leukemia diagnosis for a median duration of 6.5 years (range: 0.05–17).

Ten patients underwent allogeneic HSCT for SCD prior to leukemia. Conditioning regimens included alkylating agents and/or total body irradiation in all cases, four from matched sibling donors and six from haploidentical donors. The median interval between HSCT and leukemia diagnosis was 2.5 years (range: 0.26–7). In all patients, the leukemia arose from host-derived cells, and previous graft failure was documented in five patients (data not mentioned in the other five cases).

Two AML cases occurred after gene therapy with lentiviral vector-based LentiGlobin after myeloablation with an alkylating agent [42,44].

Overall, acute leukemias in SCD patients had a poor prognosis, and 24 patients died (60% of the 40 with outcome reported), with overall survival ranging from a few days to more than 2.5 years (median: 7 months).

Case reports also describe chronic myeloid leukemia (CML) and chronic lymphocytic leukemia (CLL) in SCD patients. An abstract derived from the United States National Inpatient Sample database identified 60 patients hospitalized with both CML and SCD, with a median age of 40.5 years, which is younger than the median age of CML diagnosis in the general population (64 years) [48].

In sum, in addition to an increased risk of AML, SCD-associated AML has some specificities with frequent underlying MDS and unfavorable karyotype. This suggests some unique features in the underlying mechanisms.

## 4. Potential Biological Mechanisms Underlying the Increased Cancer Risk

Multiple mechanisms could contribute to carcinogenesis and cancer specificities in patients with SCD (Figure 1).

### 4.1. Chronic Inflammation and Oxidative Stress

Chronic inflammation and oxidative stress are central features of SCD, contributing to persistent organ damage and systemic cellular stress [49].

SCD is associated with chronic inflammation with elevated type 1 cytokines and activation of key cellular mediators of inflammation, such as neutrophils, monocytes, endothelial cells, and platelets [50]. This inflammation is driven by multiple mechanisms, including activation of coagulation and endothelial cells by altered RBC, release of RBC components, including HbS that can act as a damage-associated molecular pattern, and ischemia–reperfusion cycles [50,51]. Persistent inflammation can promote sustained DNA damage [52], and inflammasomes may participate in this process by modulating cell proliferation, inhibiting apoptosis, and possibly contributing to impaired anti-tumor immune surveillance [53,54,55].

Patients with SCD are also known to experience chronic oxidative stress [54,56], with increased generation of reactive oxygen species (ROS) [57]. ROSs contribute to carcinogenesis through multiple pathways, including genomic instability, insufficient control of cell growth, cell death escape, angiogenesis, epithelial–mesenchymal transition, tumor microenvironment perturbation, and ferroptosis, an iron-dependent cell death pathway [58,59]. The long-term oncogenic consequence of oxidative stress in SCD remains to be assessed. It is plausible that ROS-mediated injury may contribute to genomic instability identified in lymphocytes of patients with SCD [60] and could foster a microenvironment conducive to oncogenic transformation.

### 4.2. Increased Hematopoietic Turnover and Ineffective Erythropoiesis

SCD-associated hemolysis results in an increased turnover of blood cells and hematopoietic hyperplasia (Figure 2). By removing hematopoietic stem and progenitor cells (HSPCs) from dormancy, this increased turnover may, by itself, result in DNA damage [61].

One surprising discovery was that despite this hematopoietic hyperplasia, SCD was also associated with ineffective erythropoiesis [3]. In effect, HbS not only induces hemolysis of RBCs but also that of erythroblasts in the bone marrow [62]. Chronic inflammation present in SCD may further potentiate ineffective erythropoiesis in a GATA1-dependent mechanism [63]. Ineffective erythropoiesis has been shown to favor the selection of clones carrying a selective advantage, such as those with a pre-leukemic somatic variant [64], notably due to local inflammation [65]. In SCD, RBC precursors with high fetal hemoglobin content have been shown to confer a proliferation advantage [62]. Whether the specific bone marrow environment in SCD also favors the occurrence and/or the expansion of clones carrying malignancy-associated somatic variants remains to be clarified [66,67]. Emerging evidence suggests that ineffective erythropoiesis in thalassemia may similarly favor somatic mutations, highlighting a potential shared mechanism between these disorders [68].

### 4.3. Premature Hematopoietic Aging

Premature cellular senescence and accelerated biological aging have been increasingly recognized as potential contributors to leukemogenesis in SCD. Chronic oxidative stress, systemic inflammation, repeated bone marrow injury, and the sustained high turnover of HSPCs in SCD can promote early exhaustion and senescence of the hematopoietic compartment [69]. Senescent cells may adopt a senescence-associated secretory phenotype (SASP), characterized by the release of pro-inflammatory and pro-fibrotic mediators that disrupt the marrow microenvironment and facilitate malignant transformation [70].

More broadly, multiple pieces of evidence identified hallmarks of accelerated aging in SCD [4]. Aging is associated with many molecular and cellular phenomena, including cellular senescence, chronic inflammation, telomere shortening, mitochondrial dysfunction contributing to increased oxidative stress, and epigenetic changes [71]. The speed of onset of these changes is influenced by genetic and environmental factors [72]. SCD has been associated with several phenomena involved in aging, such as increased oxidative stress and chronic inflammation [54,55,57]. Several markers of aging have been found more frequently in SCD, such as telomere shortening [73,74], a decrease in ceramides (sphingolipids whose serum concentration decreases with age), and an increase in polyubiquitinil proteins [75,76]. Epigenetic clock analyses have found accelerated aging signatures with the most recent clock models [77]. In addition, several organ dysfunctions related to SCD (such as cardiac and kidney damage) have a pathogenesis similar to age-related dysfunctions [4].

These hallmarks of aging may contribute to clonal instability, impair DNA repair mechanisms, and generate selective pressures that favor the expansion of pre-leukemic clones [78]. Furthermore, aging of HSCs has been proposed as a central mechanism predisposing to hematological malignancies [79].

### 4.4. Defect in Tumoral Immunosurveillance

Functional asplenia is a well-documented feature of SCD, resulting in a reduced capacity to clear circulating abnormal or aged cells and impaired control of some infections, mainly due to encapsulated bacteria [80], and an increased risk of thrombo-embolic complications [81,82,83,84]. Humoral and cellular immune responses, notably involving IgM+ memory B cells, are impaired from the first months of life [85]. A large cohort study involving over 8000 cancer-free American veterans who had undergone splenectomy reported a significantly increased long-term risk of developing various malignancies, including hematological malignancies [86]. These findings suggest that the spleen contributes to tumor immunosurveillance and that its absence may create a permissive environment for malignant transformation.

Patients with SCD also exhibit quantitative and qualitative alterations in multiple immune cell compartments, including chronic activation and functional impairment of T cells and natural killer cells [87,88], as well as possible T regulatory cell defects [89]. B cells and invariant natural killer T cells could also be altered [88]. A recent study identified that SCD leads to alteration of chromatin conformation in CD8+ T cells, suppressing genes involved in ferroptosis and eventually weakening anti-tumor immunity and promoting tumor growth [90]. Together, these immune alterations suggest a defect in tumor immunosurveillance in SCD that is currently largely underexplored.

Microbiomes can be one of the contributing factors of altered immune surveillance in SCD. Patients with SCD have been reported to display distinct microbiome profiles compared to the general population, potentially shaped by chronic inflammation, frequent antibiotic exposure, and dietary factors [91,92]. Alterations in the gut microbiome have been implicated in carcinogenesis through immune modulation and alterations in metabolic signaling [93,94]. Modulation of microbiome composition appears as a potential intervention to limit tumor development or favor response to treatment [95]. It remains to be determined whether the microbiome alterations observed in SCD overlap with those associated with increased cancer risk.

In addition to tumor immunosurveillance, alterations of the tumor microenvironment could be involved in the increased susceptibility to cancer found in SCD. Chronic hemolysis and tissue hypoxia can promote angiogenesis and neovascularization [96]. This adaptive response may also foster a microenvironment that is permissive to tumor initiation or progression.

### 4.5. Clonal Hematopoiesis in Sickle Cell Disease

Clonal hematopoiesis (CH) is defined as the occurrence of somatic variants in HSPC, leading to the expansion of the concerned clone(s) that will eventually contribute to a substantial part of circulating blood cells [97]. The somatic variants typically occur in genes associated with myeloid malignancies, and the most frequently affected genes are *DNMT3A*, *TET2*, *ASXL1*, and *TP53* [98]. Despite CH being an age-related phenomenon [99,100], CH can occur earlier in some hematological diseases such as aplastic anemia and inherited bone marrow failure [101,102]. CH has been linked to chronic inflammation and several unfavorable outcomes such as increased risk of hematological malignancies, cardiovascular disease, and overall mortality [100,103,104]. In SCD, the chronic inflammatory state, oxidative stress, and increased hematopoietic turnover could theoretically increase CH occurrence [105]. One study found that CH was detected earlier in patients with SCD and more frequently in all age groups compared to individuals without SCD [106], but these results were not confirmed by another study [107]. Of note, hydroxyurea use was not associated with an increased risk. These studies were both limited by the shallow sequencing approaches used, which may greatly impact the detection of CH. Nevertheless, the first study of CH in SCD identified and confirmed the presence of CH, including high-risk clones with somatic variants in *TP53* in young individuals (<40), an age at which no CH is typically detected with the sequencing method used [106]. The presence of somatic variants in *TP53* was also reported in SCD patients who later developed acute leukemia following HSCT failure or gene therapy, with pre-existing mutations sometimes identified at baseline [40,42,106,108,109]. Finally, single-colony sequencing of HSPCs of patients with SCD identified a number of somatic variants higher than expected in the general population [110]. Thus, several pieces of evidence suggest that CH is likely increased in patients with SCD, but data are currently discordant and require a large-scale study with sensitive and accurate sequencing for a faithful assessment. Regardless of whether an increased risk exists, the presence of CH, especially high-risk clones, suggests that screening before potentially curative therapies could improve risk stratification [111].

In sum, several mechanisms could be involved in the baseline increased risk of malignancy associated with SCD. Nevertheless, which mechanism(s) are actually involved and their respective contributions are largely unknown. In addition to SCD-related mechanisms, this risk could be modulated by the various therapies that can be used in SCD.

## 5. Impact of Therapeutic Exposures on the Risk of Malignant Transformation

### 5.1. Does Hydroxyurea Have an Effect, and if So, Is It Protective or Damaging?

Hydroxyurea has significantly improved survival and quality of life in patients with SCD, particularly in high-resource settings. Its therapeutic benefit is primarily mediated by an increase in fetal hemoglobin levels, which inhibits hemoglobin S polymerization, and by improved microvascular blood flow through the modulation of adhesion molecule expression on RBCs, white blood cells, platelets, and endothelium [112,113]. Several randomized controlled trials have demonstrated hydroxyurea’s efficacy notably in reducing vaso-occlusive crises and mortality, with a favorable safety profile even after long-term exposure [114,115,116].

As a ribonucleotide reductase inhibitor that suppresses DNA synthesis, hydroxyurea has raised theoretical concerns regarding genotoxicity and leukemogenic potential (Figure 3). Studies have observed increased DNA damage in peripheral blood leukocytes among hydroxyurea-treated patients, which was associated with higher cumulative doses, treatment duration, and specific HBB*S haplotypes [117,118,119]. In vitro and mouse models found an increased number of structural DNA rearrangements upon hydroxyurea exposure [120,121]. A recent work identified a hydroxyurea-specific mutational signature in HSPCs but not an increase in the number of DNA mutations [110]. Collectively, these data suggest the existence of genomic effects associated with hydroxyurea treatment. However, hydroxyurea could have an opposite effect and be protective against the increased risk of CH and leukemia in SCD. By reducing bone marrow activity, chronic inflammation, and oxidative stress, hydroxyurea could reduce the risk for CH and leukemia. Whether the overall balance of hydroxyurea use will be an increased or reduced risk remains to be studied, but the current lack of signal toward increased risk is reassuring. Indeed, molecular signals raising suspicion toward an increased risk have not translated into a clear clinical risk: large prospective cohorts and long-term follow-up studies—including ESCORT-HU—have not reported an increased incidence of hematological malignancies in patients receiving hydroxyurea [122,123,124,125,126].

Furthermore, evidence from patients with chronic myeloproliferative disorders, in whom hydroxyurea is also widely used, has not confirmed an elevated leukemic risk [127]. It is possible that the risk of leukemia in SCD stems more from disease-specific factors rather than from hydroxyurea therapy itself.

Therefore, the benefit–risk profile of hydroxyurea remains favorable, and its continued use is recommended in both pediatric and adult patients, with appropriate long-term monitoring [128].

### 5.2. Impact of Chronic Transfusions and Iron Overload

Chronic transfusion therapy remains a cornerstone in the management of several complications of SCD, notably for patients with frequent vaso-occlusive crises or at high risk of stroke [129]. While this approach effectively reduces sickling complications, it may be associated with iron overload. Excess of iron in tissues results in perturbations in redox balance and leads to increased oxidative stress [130], which can contribute to carcinogenesis [58,59]. In addition to ROS induction, iron overload may damage organs through fibrosis and alter the function of several immune cells [131]. Iron overload could also trigger ferroptosis, contributing to epithelial–mesenchymal transition [132]. This phenomenon is central in the development of several solid tumors and subsequent metastases, but as these tumors are not the most frequently reported in individuals with SCD, it is likely not a major mechanism in this setting. Focusing on the hematopoietic system, iron overload has been shown to reduce HSPC function and renewal, mainly due to increased oxidative stress [133,134]. Although one may speculate that such an effect may favor the selection of clones carrying a somatic variant, this remains to be demonstrated in the context of SCD. Further studies are warranted to clarify whether iron overload contributes to the increased risk of malignancies in individuals with SCD. In the Californian cohort, patients with severe SCD (≥3 visits per year) had a more pronounced risk of leukemia than others [12]. However, whether this difference was driven by disease- and/or therapeutic-specific mechanisms is unknown.

### 5.3. Hematopoietic Stem Cell Transplantation and Intensive Therapeutic Regimens

In contrast to life-long supportive care, allogenic HSCT offers a potentially curative option for patients with SCD, though it may be followed by severe complications. While one study reported that HSCT for SCD does not appear to increase the risk of acute leukemia compared to non-transplant patients [135], conditioning regimens involving alkylating agents and ionizing radiation remain a concern given their carcinogenic effect demonstrated in other settings [136,137,138]. The relative rarity of SCD-associated leukemia requires very large cohorts to assess whether an increased risk is present. In the cases reported, the latency periods between transplantation and leukemia onset in reported cases fall within the range observed in therapy-related myeloid neoplasms following chemotherapy or radiation in other contexts, typically between 1 and 5 years, even if the risk remains even >10 years after initial treatment for a hematological malignancy [139]. Moreover, several of these cases exhibit high-risk cytogenetic and molecular abnormalities as commonly associated with therapy-related myeloid neoplasms, such as monosomy 7, *TP53*, or *RUNX1* mutations [40,44,108].

After HSCT for SCD, myeloid malignancies have been reported almost exclusively in patients who experienced graft failure [36,40]. In such cases, the hematopoietic stress associated with reconstitution increases proliferative pressure, potentially accelerating the acquisition of leukemogenic mutations as demonstrated in HSCT in other settings than SCD [140]. *TP53* somatic variants have been detected prior to HSCT in several patients with SCD and shown to expand over time until overt therapy-related myeloid malignancy is diagnosed [40]. The expansion of these clones has been particularly linked to chemotherapy exposure [108]. Clones harboring *TP53* mutations may also exhibit resistance to cytotoxic therapies, enabling their preferential expansion after conditioning and contributing to the development of AML following graft rejection.

Gene therapies also hold promise as a curative approach, yet emerging reports have raised concerns regarding leukemogenesis. In the largest trial of lentiviral vector-mediated β-globin gene therapy for SCD, two adult patients developed AML [42,44,141,142]. Both cases shared similar high-risk cytogenetic and molecular features, including monosomy 7 and mutations in *RUNX1* and *PTPN11*, which were absent in pre-treatment bone marrow samples. The first case was attributed to busulfan conditioning [42], whereas in the second, the presence of the vector in leukemia blasts indicated clonal origin from a transduced hematopoietic stem cell [44]. However, multiple lines of evidence argued against insertional mutagenesis as the primary driver of leukemogenesis [141].

## 6. Conclusions, Perspectives, and Clinical Implications

Available data support specificities in leukemia occurring in patients with SCD. Most data suggest the presence of an increased risk of leukemia in SCD, although definitive data are still lacking. SCD-associated leukemias are frequently AML and display frequent high-risk, unfavorable features with poor outcomes that are reminiscent of therapy-related myeloid neoplasms. Current evidence suggests SCD itself could have a leukemogenic potential. Whether SCD influences the occurrence or characteristics of other hematological malignancies and solid tumors is only scarcely studied, and more investigations are required to formulate an informed opinion at this point.

Multiple mechanisms may contribute to the association between SCD and leukemia, and the specificities of SCD-associated leukemias, including chronic inflammation, oxidative stress, ineffective erythropoiesis, increased hematopoietic turnover, aging, defect in immunosurveillance, and CH. Whether they are causally involved in SCD-associated leukemias is currently poorly known, and future investigations are required to decipher their respective contribution. Most of these factors, including CH [143], may favor both hematological and solid tumor development. The predominance of AML in patients with SCD suggests myeloid-specific effects or susceptibility in SCD.

The contribution of treatments as aggravating factors or opportunities to reduce malignancy risk is a major point that is largely unknown. The increased risk associated with HSCT, partly related to conditioning that is also used in gene therapy, is somewhat expected given the knowledge gained from other diseases. The impact of hydroxyurea and chronic transfusion is currently unknown, but no unfavorable signal has emerged. As such, current data does not suggest that SCD-associated leukemia risk should be considered in the treatment choice. Despite being historically underfunded [144], many drugs are now under study in SCD and some were recently approved [145], and their effect on leukemia risk remains to be studied. Moreover, these uncertain, non-quantifiable risks must be put in perspective with the robust cumulative evidence for hydroxyurea and blood transfusions as life-supporting and life-prolonging therapeutic interventions.

From a clinical point of view, our review highlights several practical points. First, it underscores the need for long-term oncological follow-up studies for patients with sickle cell disease. Establishing international registries and collaborative research networks will be critical to gather sufficient data and develop evidence-based guidelines. The current data does not support modifying the treatment decision based on SCD-associated leukemia risk, especially given the overwhelming benefits carried by the treatments. There is no data suggesting a benefit in systematic screening for CH in individuals with SCD. The decision to search for CH before HSCT or gene therapy is more challenging. CH could increase the risk of leukemia in case of HSCT failure or after HSCT, especially for high-risk CH, such as those driven by *TP53* variants, which have been shown as a risk factor for post-HSCT leukemia [146]. Thus, identification of CH before such an irreversible procedure could allow for the identification of patients at higher risk of leukemia. But whether the presence of CH impacts the clinical decision—initially driven by the severity of SCD and the benefit demonstrated of these procedures—or the patients’ follow-up is currently unknown. Despite its unclear clinical benefit, a standardized protocol of CH assessment before and after HSCT or gene therapy is certainly warranted from a research perspective. Longitudinal studies are urgently needed to evaluate the prevalence, mutational profiles, and prognostic significance of CH in SCD [97].

## Figures and Tables

**Figure 1 jcm-14-08509-f001:**
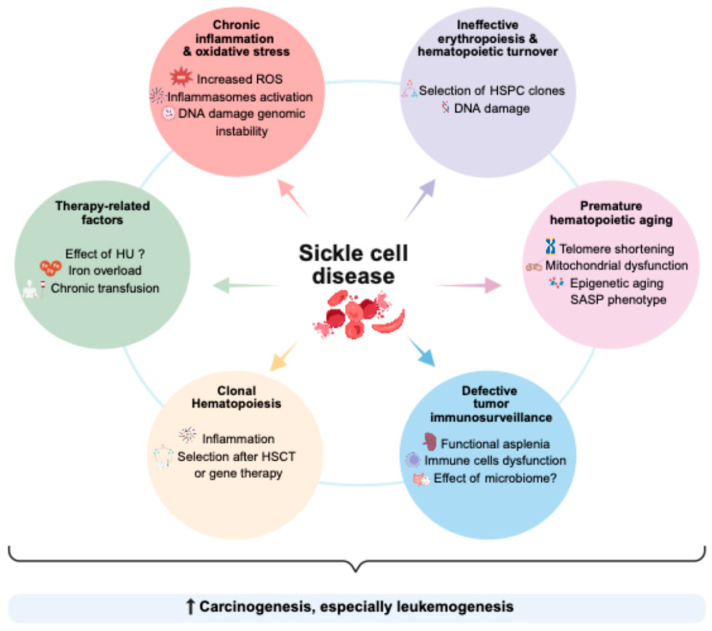
Schematic view of the multiple mechanisms that could be involved in carcinogenesis in patients with SCD. Created in BioRender. Pincez, T. (2025) Accessible online at https://BioRender.com/r56jv5d (accessed on 19 October 2025).

**Figure 2 jcm-14-08509-f002:**
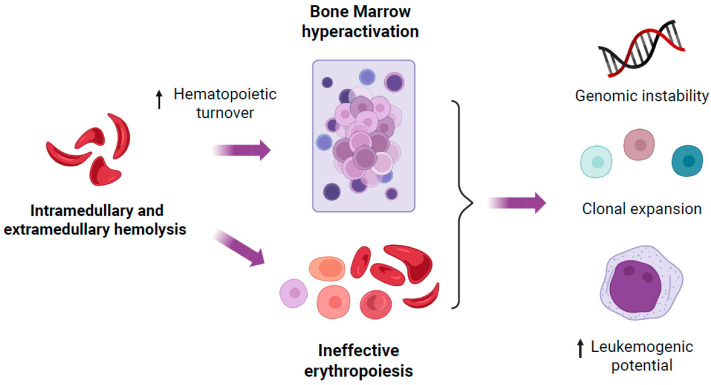
Cause and consequences of bone marrow hyperactivation and ineffective erythropoiesis in the carcinogenesis of sickle cell disease. Created in BioRender. Pincez, T. (2025) Accessible online at https://BioRender.com/4fwwmot (accessed on 19 October 2025).

**Figure 3 jcm-14-08509-f003:**
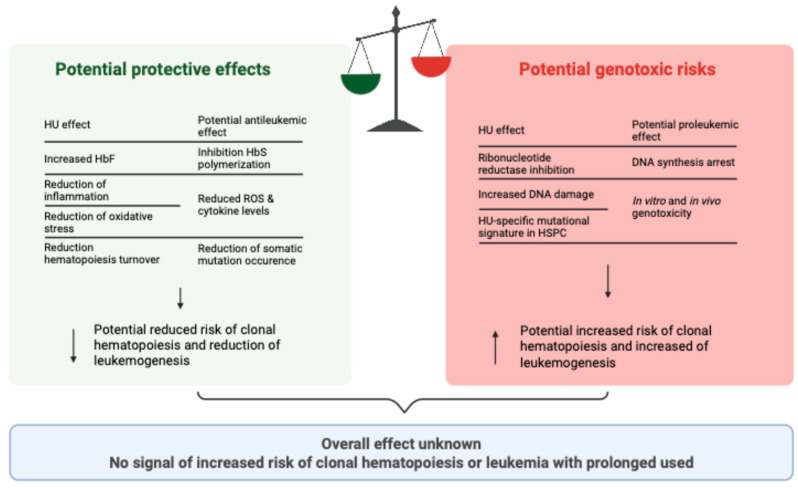
Balance between potential benefits and risks of carcinogenesis associated with hydroxyurea use in sickle cell disease. Current evidence did not identify an increased risk of clonal hematopoiesis or leukemia. Created in BioRender. Pincez, T. (2025) Accessible online at https://BioRender.com/f1sn39c (accessed on 19 October 2025).

**Table 1 jcm-14-08509-t001:** Comparative incidence ratios of malignancies in individuals with SCD across major epidemiological studies. SIR, standardized incidence ratio. RR, rate ratio.

Cancer Type	Brunson et al. (California) [12] SIR	Seminog et al. (England) [11] RR
AML	3.59	10.69
ALL	1.83	2.72
CLL	4.83	For all lymphoid leukemia
Solid tumors	0.62	-
Breast cancer	0.54	0.51
Lymphoma	1.45	Hodgkin lymphoma: 4.32 Non-Hodgkin lymphoma: 2.37

**Table 2 jcm-14-08509-t002:** Summary of clinical and biological characteristics of the 53 reported SCD-associated acute leukemias and MDS.

Variable	Category	n (%) or Median (Range)	Comment
Sex	Male/female	18 (43%)/24 (57%)	M/F ratio = 0.75
Sickle cell disease type	HbSS/HbSC/HbSβ^0^/HbSD	41 (84%)/4 (8%)/3 (6%)/1 (2%)	
Age at leukemia diagnosis	Median (min–max)	26 years (3–61)	
Prior SCD treatment			
	Hydroxyurea	24 (58%)	Median duration: 6.5 years
	Chronic transfusion	17 (32%)	
	HSCT	10 (19%)	All with relapse or graft failure
	Gene therapy	2 (4%)	Myeloablative conditioning
Leukemia type	ALL/AML/MDS/MDS-AML overlap/other	13/17/6/16/1	73% myeloid malignancies
AML subtype (ICC/WHO 2022 [15])	AML-defining ^1^	20/39 (51%)	
Unfavorable cytogenetics	−5/−7/del17/complex	20/27 (74%)	t-AML marker
Leukemia treatment	Chemotherapy alone	16 (30%)	
	HSCT	9 (17%)	
	Azacitidine/low intensity	4 (8%)	
	Supportive care/not reported	24 (45%)	
Complete remission (CR)	Yes	60%	
OS	Median (range)	7 months (4 days–2.5 years)	12-month OS: 37.5%

^1^ AML-defining: AML with MDS-related changes, AML with PML::RARA, and AML TP53-mutated and therapy-related.

**Table 3 jcm-14-08509-t003:** Acute leukemia characteristics and outcome of the 53 patients with SCD-associated acute leukemias and MDS.

Patient/Age/Sex	Type of Leukemia/MDS	Treatment	Outcome (Cause of Death)	Reference
6/F	ALL	Chemo	CR, OS: 17 months (viremia)	[16]
7/F	AML	None	OS: 4 days	[17]
27/F	MDS/AML4	None	OS: 3 days (ARDS)	[18]
8/F	AML2	HSCT	OS: 16+ months	[19]
4/F	ALL null (del9p13)	NA	NA	[20]
43/M	MDS/AML1 (−3, t13;17, t3;5, 5q−, −7, +8)	Chemo	OS: 1 month (hemorrhage)	[21]
22/M	ALL	Chemo	CR, OS: 10 months (progression)	[22]
14/M	ALL (CD10+, CD19+, CD22+, DR+, TdT+)	Chemo	CR, OS: 2.5+ years	[23]
10/F	Ph+ ALL	Chemo	CR, OS: 12+ months	[24]
27/F	MDS/AML	NA	NA	[25]
42/F	MDS/AML (−5, −7, del17)	Chemo	OS: 13 months	[26]
25/F	AML1 (normal karyotype)	Chemo	CR, OS: NA (aspergillosis)	[27]
14/F	ALL	NA	Alive	[10]
5/NA	ALL	NA	Alive	
7/NA	ALL	NA	Alive	
8/NA	AML	NA	Alive	
8/NA	ALL	NA	Alive	
17/NA	ALL	NA	Alive	
61/NA	ALL	NA	Dead	
20/NA	AML	NA	Dead	
21/F	AML3v	ATRA + Chemo	CR	[28]
33/M	MDS/AML6 (Abn5q, del7q, −15, −22, −Y, mar5)	Chemo + Allo HSCT	CR, relapse at 4 m, OS: 9 m	[29]
41/M	MDS/AML (Abn5, del7, −17)	Chemo	OS: 3 m (sepsis)	[30]
55/M	MDS/AML (5q−, 7q−, del17p)	NA	NA	[31]
49/M	MDS/AML6 (del17p, del5q, monosomy 20, BM fibrosis)	Chemo	OS: 3 w (CNS involvement)	[32]
25/F	AML3	NA	NA	[33]
19/M	AML2	NA	NA	
31/F	MDS/AML (5q−, add5p, −7, t2;5, TP53+, NRas+)	Azacitidine	OS: 12 m (sepsis)	[34]
59/F	MDS (del4, 5q−, 7q−, −15, −16, TP53+)	Decitabine	OS: 2 m (AML progression)	[35]
27/M	MDS/AML (11q23, +3, +19, +21, KMT2A+)	Chemo + Allo HSCT	OS: 7 m	
37/F	MDS (del1, del5, t3;6, −17, +3, TP53+)	Lenalidomide + prednisone	OS: 5+ m	
34/M	MDS (7q22, del20, −2, Inv9)	Matched sibling HSCT	OS: 21+ m	
19/NA	AML	NA	NA	* [36]
37/NA	MDS	NA	NA	
32/NA	AML	NA	NA	
37/NA	MDS	NA	NA	
26/F	MDS/AML (5q−, +8, del17, TP53 del)	Chemo	OS: 4 m	[37]
15/M	AL mixed lineage	None	Death before treatment	[38]
21/F	ALL	None	Discharged day 5 CR after 2 lines	
15/M	AML4	Chemo	CR, death 4 w later	
3/M	AML	None	Discharged after diagnosis	
15/F	AML5	Chemo	OS: 2 m (sepsis)	
29/F	AML6 (5q−)	Chemo	OS: a few months (AML progression)	[39]
39/M	MDS/AML7 (complex cytogenetics, TP53+, BM fibrosis)	Decitabine + Azacitidine	OS: 12 m (pulmonary hypertension)	[40]
39/M	MDS/AML (complex cytogenetics, TP53+)	Haplo HSCT	OS: 7 m (intracranial hemorrhage)	
49/F	MDS/AML (7q−, BM fibrosis)	NA	NA	
14/F	AML CNS+ (FLT3-ITD+)	Chemo + sorafenib + Haplo HSCT	CR, OS: 8+ m	[41]
42/M	MDS/AML (−7, 19p Abn, RUNX1+, KRAS+, PTPN11+)	Azacitidine, Decitabine, Chemo, Haplo HSCT	CR after Haplo, OS: 6+ m	[42]
19/M	Ph+ ALL	Chemo + imatinib	OS: 6 m (meningoencephalitis)	[43]
31/F	AML0 (−7, 11p−, WT1+, RUNX1+, PTPN11+)	Chemo + Haplo HSCT	CR (MRD+), OS: 12 m (AML progression)	[44]
40/M	MDS (complex cytogenetics, 5q−, 3p, 7p, −16, −7, −18)	None	OS: 3 m (severe cytopenia)	[45]
27/F	MDS/AML (−3, t5;7, −7, del12, −22, TP53+)	Vyxeos, MEC, HSCT	CR, MRD− after HSCT, OS: 12+ m	[46]
48/F	AML3	Chemo + ATRA + arsenic trioxide	CR, MRD−	[47]

NA, non applicable or not available; Abn, abnormality; AL, acute leukemia; ALL, acute lymphoblastic leukemia; Allo, allogeneic; AML, acute myeloid leukemia; ARDS, acute respiratory distress syndrome; ATRA, all-trans retinoic acid; BM, bone marrow; Chemo, intensive chemotherapy; CNS, central nervous system; CR, complete remission; F, female; Haplo, haploidentical; HSCT, hematopoietic stem cell transplantation; M, male; MEC, chemotherapy combining mitoxantrone, etoposide, and cytarabine; MDS, myelodysplastic syndrome; MRD, measurable residual disease; OS, overall survival; Ph+, Philadelphia chromosome-positive. * This reference is based on registry data. The patients may therefore overlap with others reported in the table.

**Table 4 jcm-14-08509-t004:** SCD characteristics of the 53 patients identified in the literature.

Patient/Diagnosis/Origin	Age at Diagnosis	Treatment	Clinical Features	Reference
SS/Afr.Am	NA	No HU	NA	[16]
SS/NA	Infancy	Transfusions	NA	[17]
SS/NA	NA	Transfusions	Hemosiderosis	[18]
SS/Afr.Am	Infancy	NA	NA	[19]
SS/NA	At birth	NA	NA	[20]
SC/Afr.Am	NA	NA	Aseptic necrosis humeral head	[21]
SS/Nigerian	NA	NA	NA	[22]
SS/Afr.Am	Infancy	No HU	VOC	[23]
SS/NA	Infancy	HU (1.5 months)	VOC (3–7/year)	[24]
SS/NA	NA	HU (8 years)	VOC	[25]
SS/NA	NA	HU (6 years)	NA	[26]
SS/Saudi	NA	HU (2 years)	VOC (6/year), hepatitis C	[27]
SS/NA	Infancy	HU (3 months)	NA	[10]
SS/NA	Infancy	No HU	NA	
SS/NA	Infancy	No HU	NA	
SS/NA	Infancy	No HU, HSCT	NA	
SS/NA	Infancy	No HU	NA	
SS/NA	NA	No HU	NA	
SS/NA	NA	No HU	NA	
SS/NA	NA	No HU	NA	
SS/NA	NA	HU (8 years)	VOC, osteonecrosis, ACS	[28]
SS/Afr.Am	NA	HU (5 years), transfusions	VOC, priapism, ACS	[29]
SS/Afr.Am	21	Exchange transfusions, HU (15 years)	VOC (14 to 3/year)	[30]
SS/Jamaican	NA	No HU	Pulmonary hypertension	[31]
SS/NA	NA	HU (14 years), transfusions	VOC, hip necrosis, retinopathy, stroke, iron overload	[32]
SS/Indian	NA	Transfusions, HU	NA	[33]
SS/Indian	NA	HU	NA	
SS/Afr.Am	Childhood	HU (5 years), Haplo HSCT (8 months)	VOC	[34]
SC/NA	NA	HU, exchange transfusions	HHV8	[35]
SS/NA	NA	Exchange transfusions	VOC, myocardial infarction, HIV+	
SS/NA	Infancy	Exchange transfusions	VOC	
Sβ^0^/NA	NA	Exchange transfusions, HU (9 years) matched HSCT (7 years)	VOC, priapism, arterial aneurysm, intracranial bleeding	
NA/NA	NA	Haplo HSCT (3.6 years)	NA	* [36]
NA/NA	NA	Haplo HSCT (9 months)	NA	
NA/NA	NA	Haplo HSCT (1 year)	NA	
NA/NA	NA	Matched sibling HSCT (2.6 years)	NA	
SS/Afr.Am	Childhood	Transfusion/exchange HU (2 years)	VOC, pulmonary fibrosis, pneumonia, hip necrosis, peritonitis	[37]
SS/Nigerian	2 years	No HU, transfusion	VOC (2/year)	[38]
SS/Nigerian	4 years	No HU	VOC (1/year)	
SC/Nigerian	Childhood	No HU	VOC (once in 2–3 years)	
SC/Nigerian	NA	No HU	None	
SS/Nigerian	NA	No HU, transfusions	NA	
SS/NA	NA	HU (5 years)	VOC	[39]
SS/NA	NA	HU, Haplo HSCT (2 years)	Stroke, CRI, VOC	[40]
SS/NA	NA	HU, sibling HSCT (2.5 years)	VOC	
SS/NA	NA	HU, Haplo HSCT (5 years)	Diastolic dysfunction, ESRD, pulmonary hypertension	
Sβ^0^/Haitian	At birth	HU (9 years)	VOC	[41]
SS/NA	NA	HU (8 years), gene therapy	VOC, iron overload, hypertension	[42]
SS/Nigerian	At 1 year	Transfusions, no HU	VOC (>4/year)	[43]
SS/NA	NA	HU (6 years), gene therapy (LentiGlobin) (5.5 years)	VOC, hip necrosis, deep-vein thrombosis	[44]
SS/NA	NA	HU (17 years), exchange transfusions	VOC, priapism, pulmonary hypertension	[45]
Sβ^0^/African	Childhood	HU (7 years), exchange transfusions	VOC, ACS, retinopathy, cholelithiasis, COVID 19	[46]
SD/NA	NA	No HU, transfusions	Hemolytic crisis, bone osteonecrosis and sclerosis, ACS	[47]

ACS, acute chest syndrome; Afr.Am, African American; CRI, chronic renal insufficiency; ESRD, end-stage renal disease; Haplo, haploidentical; HIV, human immunodeficiency virus; HSCT, hematopoietic stem cell transplantation; HU, hydroxyurea; LentiGlobin, gene therapy consisting of autologous hematopoietic stem and progenitor cells transduced with the BB305 lentiviral vector encoding the βA−T87Q-globin gene designed to produced anti-sickling hemoglobin (HbAY87Q); NA, not available; VOC, vaso-occlusive crisis. * This reference is based on registry data. The patients may therefore overlap with others reported in the table.

## Data Availability

No new data were created or analyzed in this study.

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
