# Peer review of "When Blood Disorders Meet Cancer: Uncovering the Oncogenic Landscape of Sickle Cell Disease"

_jcm, 2025, doi:10.3390/jcm14238509_

Round 1
Reviewer 1 Report
Comments and Suggestions for Authors
The review “When Blood Disorders Meet Cancer: Uncovering the Oncogenic Landscape of Sickle Cell Disease” analyzes the link between sickle cell disease and malignancy development, especially leukemia. Recommendations:
- The introduction should emphasize the clinical impact and novelty of exploring cancer risk in SCD, highlighting how improved survival has unmasked oncologic complications.
- The epidemiological section needs better structure and summarization. Key findings from large cohorts should be compared more explicitly to outline consistent trends and discrepancies.
- In Table 1 and 2, include only essential variables and citation numbers—avoid excessive details that obscure interpretation.
- The statistical interpretation of reported cases and incidence rates should be strengthened. Consider summarizing frequency and risk ratios in a compact figure or meta-analytic table.
- The figures illustrating mechanisms are conceptually strong but not professionally designed. Redraw them to improve clarity and visual hierarchy.
- The mechanistic section is well referenced but overly descriptive. Simplify and better connect these processes to leukemogenesis in SCD.
- The role of hydroxyurea is well presented; however, the balance between its theoretical genotoxicity and clinical safety should be summarized in a concise comparative table.
- The discussion should highlight the lack of causal data and suggest directions for future prospective studies, including standardized CH screening protocols before HSCT or gene therapy.
- Discuss the importance of the microbiota in hematological pathologies, especially in childhood, focusing on disorders that could progress to blood diseases, including those of malignant origin, but not exclusively – recommended reference: 10.3390/children12020166.
- The conclusion is comprehensive but too long.
Author Response
The review “When Blood Disorders Meet Cancer: Uncovering the Oncogenic Landscape of Sickle Cell Disease” analyzes the link between sickle cell disease and malignancy development, especially leukemia. Recommendations:
Q1. The introduction should emphasize the clinical impact and novelty of exploring cancer risk in SCD, highlighting how improved survival has unmasked oncologic complications.
R1. We thank the reviewer for this suggestion. We have clarified this point in the introduction by adding or modifying the following sentences:
"The advances in care made in the last decades have markedly improved life expectancy, thereby unmasking age-related complications, such as bone disease, pulmonary hypertension, neurocognitive impairment [7,8], but also malignancies. This report of malignancies, mainly leukemia, was somewhat surprising. Although the coincidental occurrence of leukemia in individuals with SCD was expected, clinical features of these leukemia and epidemiological data suggested that SCD could be an independent risk factor for leukemia occurrence."
"The potential cancer risk associated with SCD is poorly known and this novel research field uncovered an important aspect of this disease. This cancer risk could be modified by the treatment received and could by itself impact treatment consideration."
Q2. The epidemiological section needs better structure and summarization. Key findings from large cohorts should be compared more explicitly to outline consistent trends and discrepancies.
R2. We have modified the section for better clarity and summarized the results of the two large epidemiological cohorts in the last paragraph. The following sentences were modified or added:
"Conversely, a subsequent multicenter study including 16,613 patients and two epide-miological studies suggested that the risk of solid tumor was only marginally increased and that hematological malignancies and especially acute leukemias were notably overrepresented in individuals with SCD [10–12]. The two epidemiological studies used population databases to compare individuals with and without SCD [13]."
"Thus, the two epidemiological studies are concordant toward an increased risk of leukemia, especially AML, despite it precise magnitude remains to be clarified. While both studies found a reduction in breast cancer, data regarding other solid tumors are discrepant."
Q3. In Table 1 and 2, include only essential variables and citation numbers—avoid excessive details that obscure interpretation.
R3. We have reduced the data in the table, including removal of the references' names and number.
Q4. The statistical interpretation of reported cases and incidence rates should be strengthened. Consider summarizing frequency and risk ratios in a compact figure or meta-analytic table.
R4. As suggested, we have added a table to summarize and compare the results of the two large epidemiological studies.
Q5. The figures illustrating mechanisms are conceptually strong but not professionally designed. Redraw them to improve clarity and visual hierarchy.
R6. We have modified all figures and remove previous Figure 1 for a new figure. All were professionally drawn using state-of-the-art Biorender software.
Q7. The mechanistic section is well referenced but overly descriptive. Simplify and better connect these processes to leukemogenesis in SCD.
R7. We have now modified this section for conciseness and highlight the link between these processes and SCD-related leukemogenesis. The following sentences have been modified:
"SCD is associated with chronic inflammation with elevated type 1 cytokines and activation of key cellular mediators of inflammation such as neutrophils, monocytes, endothelial cells, and platelets [50]. This inflammation is driven by multiple mechanisms, including activation of coagulation and endothelial cells by altered RBC, releasing of RBC components including HbS that can act as damage associated molecular pattern, and is-chemia-reperfusion cycles [50,51]. Persistent inflammation can promote sustained DNA damage [52], and inflammasomes may participate in this process by modulating cell proliferation, inhibiting apoptosis, and possibly contributing to impaired anti-tumor immune surveillance [53–55]. "
"Patients with SCD are also known to experience chronic oxidative stress [56,57], with increased generation of reactive oxygen species (ROS) [58]. ROS contribute to carcinogenesis through multiple pathways, including genomic instability, insufficient control of cell growth, cell death escape, angiogenesis, epithelial-mesenchymal transition, tumor microenvironment perturbation, and ferroptosis, an iron-dependent cell death pathway [59,60]. The long-term oncogenic consequence of oxidative stress in SCD remains to be assessed. "
"One surprising discovery was that despite this hematopoietic hyperplasia, SCD was also associated with ineffective erythropoiesis [3]. "
"Whether the specific bone marrow environment in SCD also favors the occurrence and/or the expansion of clones carrying malignancy-associated somatic variants remains to be clarified [67,68]. Emerging evidence suggests that ineffective erythropoiesis in thalassemia may similarly favor somatic mutations, highlighting a potential shared mechanism between these disorders [69]."
"The speed of onset of these changes is influenced by genetic and environmental factors [73]."
"These hallmarks of aging may contribute to clonal instability, impair DNA repair mechanisms, and generate selective pressures that favor the expansion of pre-leukemic clones [80]. Furthermore, aging of HSCs has been proposed as a central mechanism predisposing to hematological malignancies [81]."
"Functional asplenia is a well-documented feature of SCD, resulting in a reduced capacity to clear circulating abnormal or aged cells and impaired control of some infections, mainly due to encapsulated bacteria [82], and an increased risk of thrombo-embolic complications [83–86]. Humoral and cellular immune responses notably involving IgM+ memory B cells are impaired from the first months of life [87]."
"These findings suggest that the spleen contributes to tumor immunosurveillance and that its absence may create a permissive environment for malignant transformation.
Patients with SCD also exhibit quantitative and qualitative alterations in multiple immune cell compartments, including chronic activation and functional impairment of T cells and natural killer cells [89,90], as well as possible T regulatory cells defect [91]. "
"Together, these immune alterations suggest a defect in tumor immunosurveillance in SCD that is currently largely underexplored."
"Alterations in gut microbiome have been implicated in carcinogenesis through immune modulation and alterations of the metabolic signaling [95,96]. Modulation of microbiome composition appears as a potential intervention to limits tumor development or favor response to treatment [97]."
"Despite CH is an age-related phenomenon [101,102], CH can occur earlier in some hematological diseases such as aplastic anemia and inherited bone marrow failure [103,104]. "
"One study found that CH was detected earlier in patients with SCD and more frequent in all age groups compared to individuals without SCD [108], but these results were not confirmed by another study [109]. "
Q8. The role of hydroxyurea is well presented; however, the balance between its theoretical genotoxicity and clinical safety should be summarized in a concise comparative table.
R8. We have now modified the Figure 3 to incorporate a table with theoretical genotoxicity and potential benefits.
Q9. The discussion should highlight the lack of causal data and suggest directions for future prospective studies, including standardized CH screening protocols before HSCT or gene therapy.
R9. As suggested, we have highlighted the paucity of causal data in SCD related to leukemia risk and suggested the future directions in CH studies. The following sentences were modified or added:
"Multiple mechanisms may contribute to the association between SCD and leukemia and the specificities of SCD-associated leukemias, including chronic inflammation, oxi-dative stress, ineffective erythropoiesis, increased hematopoietic turnover, aging, defect in immunosurveillance, and CH. Whether they are causally involved in SCD-associated leukemias is currently poorly known and future investigations are required to decipher their respective contribution."
"Despite being historically underfunded [146], many drugs are now under study in SCD and some were recently approved [147], and their effect on leukemia risk remains to be studied."
"From a clinical point of view, our review highlights several practical points. First, it underscores the need for long-term oncological follow-up studies for patients with sickle cell disease. Establishing international registries and collaborative research networks will be critical to gather sufficient data and develop evidence-based guidelines."
"Thus, identification of CH before such irreversible procedure could allow to identify patients at higher risk of leukemia. But whether the presence of CH should impact the clinical decision —initially driven by the severity of SCD and the benefit demonstrated of these procedures— or the patients’ follow-up is currently unknown. Despite its unclear clinical benefit, a standardized protocol of CH assessment before and after HSCT or gene therapy is certainly warranted on a research perspective. Longitudinal studies are urgently needed to evaluate the prevalence, mutational profiles, and prognostic significance of CH in SCD [149]."
Q10. Discuss the importance of the microbiota in hematological pathologies, especially in childhood, focusing on disorders that could progress to blood diseases, including those of malignant origin, but not exclusively – recommended reference: 10.3390/children12020166.
R10. We have now more detailed the microbiome in the dedicated paragraph, while respecting the conciseness suggested for this section on mechanisms. We have also added the suggested reference.
"Alterations in gut microbiome have been implicated in carcinogenesis through immune modulation and alterations of the metabolic signaling [95,96]. Modulation of microbiome composition appears as a potential intervention to limits tumor development or favor response to treatment [97]."
Q11. The conclusion is comprehensive but too long.
R11. We have now shortened the section "Conclusion, perspectives, and clinical implications" while adding the points suggested in 9:
"Available data support specificities in leukemia occurring in patients with SCD. Most data suggest the presence of an increased risk of leukemia in SCD despite definitive data are still lacking. SCD-associated leukemias are frequently AML and display frequent high-risk, unfavorable features with poor outcomes that are reminiscent of therapy-related myeloid neoplasms. Current evidence suggests SCD itself could have a leukemogenic potential. Whether SCD influences the occurrence or characteristics of other hematological malignancies and solid tumors is only scarcely studied and more investigations are required to formulate an informed opinion at this point.
Multiple mechanisms may contribute to the association between SCD and leukemia and the specificities of SCD-associated leukemias, including chronic inflammation, oxidative stress, ineffective erythropoiesis, increased hematopoietic turnover, aging, defect in immunosurveillance, and CH. Whether they are causally involved in SCD-associated leukemias is currently poorly known and future investigations are required to decipher their respective contribution. Most of these factors, including CH [145], may favor both hematological and solid tumor development. The predominance of AML in patients with SCD suggest myeloid-specific effects or susceptibility in SCD.
The contribution of treatments as aggravating factors or opportunity to reduce malignancy risk is a major point that is largely unknown. The increased risk associated with HSCT, partly related to conditioning that is also used in gene therapy, is somewhat expected given the knowledge gained from other diseases. The impact of hydroxyurea and chronic transfusion is currently unknown but no unfavorable signal has emerged. As such, current data does not suggest that SCD-associated leukemia risk should be considered in the treatment choice. Despite being historically underfunded [146], many drugs are now under study in SCD and some were recently approved [147], and their effect on leukemia risk remains to be studied. Moreover, these uncertain, non-quantifiable risks must be put in perspective with the robust cumulative evidence for hydroxyurea and blood transfusions as life-supporting and life-prolonging therapeutic interventions.
From a clinical point of view, our review highlights several practical points. First, it underscores the need for long-term oncological follow-up studies for patients with sickle cell disease. Establishing international registries and collaborative research networks will be critical to gather sufficient data and develop evidence-based guidelines. The current data does not support to modify the treatment decision based on SCD-associated leukemia risk, especially given the overwhelming benefits carried by the treatments. There is no data suggesting a benefit in systematic screening for CH in individuals with SCD. The decision to search for CH before HSCT or gene therapy is more challenging. CH could increase the risk of leukemia in case of HSCT failure or after HSCT, especially for high-risk CH such as those driven by TP53 variants which have been shown as a risk factor for post-HSCT leukemia [148]. Thus, identification of CH before such irreversible procedure could allow to identify patients at higher risk of leukemia. But whether the presence of CH should impact the clinical decision —initially driven by the severity of SCD and the benefit demonstrated of these procedures— or the patients’ follow-up is currently unknown. Despite its unclear clinical benefit, a standardized protocol of CH assessment before and after HSCT or gene therapy is certainly warranted on a research perspective. Longitudinal studies are urgently needed to evaluate the prevalence, mutational profiles, and prognostic significance of CH in SCD [149]."
Reviewer 2 Report
Comments and Suggestions for Authors
Review Report
Title: “When Blood Disorders Meet Cancer: Uncovering the Oncogenic Landscape of Sickle Cell Disease”
Authors: Elise Casadessus, Manon Saby, Stéphanie Forté, Yves Pastore, Vincent-Philippe Lavallée, Thomas Pincez
Comments:
In their paper “When Blood Disorders Meet Cancer: Uncovering the Oncogenic Landscape of Sickle Cell Disease” Casadessus and colleagues aimed at incorporating current knowledge on the oncogenic landscape of Sickle cell disease (SCD), with focusing on the incidence and characteristics of leukemia in this population, whether arising spontaneously or following therapeutic interventions, and at examining the potential biological mechanisms linking SCD to increased cancer susceptibility
The article is a well-designed review, investigating the effect of SCD on carcinogenesis with data supporting specificities in SCD- associated acute myeloid leukemia (AML). They cited many epidemiological studies suggesting that individuals with SCD may have an increased risk of cancer, particularly hematological malignancies such as acute leukemias. In addition, they summarized in tables the clinical and biological characteristics of SCD-associated acute leukemias and myelodysplastic syndrome (MDS), SCD characteristics; and Acute leukemia characteristics and outcome using published data of patients with SCD-associated acute leukemias and MDS. They found that in addition to an increased risk of AML, SCD-associated AML has some specificities with frequent underlying MDS and unfavorable karyotype. And suggested some unique features in the underlying mechanisms. Furthermore, they suggested multiples mechanisms could contribute to carcinogenesis and cancer specificities in patients with SCD and highlighted these potential biological mechanisms in clear and understandable figures. In addition, they suggested that increased risk of malignancy associated with SCD could be modulated by the various therapies used in SCD and explained the impact of therapeutic exposures on the risk of malignant transformation using a figure summarizing the balance between potential benefits and risks on carcinogenesis associated with hydroxyurea use in sickle cell disease.
Taken together, the review is well conducted and adds substantial new knowledge regarding the presence of an increased risk of leukemia in patients with SCD. The findings of this review underscored the need for long-term oncological follow-up studies for patients with sickle cell disease. Consequently, this could be beneficial for patients with SCD by identifying patients at high risk of leukemia and improve the outcome of SCD-associated AML.
The authors have satisfactorily addressed all points regarding the association between SCD and malignancies. No modifications are necessary to be added to the manuscript.

Author Response
We thank very much the reviewer for the comments.
Round 2
Reviewer 1 Report
Comments and Suggestions for Authors
The authors made all recommended changes.